# Characterising the mechanics of cell–cell adhesion in plants

Asal Atakhani[iD], Léa Bogdziewiez[iD] and Stéphane Verger[iD]

Department of Forest Genetics and Plant Physiology, Umeå Plant Science Centre, Swedish University of Agricultural Sciences, Umeå, Sweden

adhesion strength; cell–cell adhesion; multicellularity; plant; single cell; tissue.

**Author for correspondence:**
S. Verger,
E-mail: stephane.verger@slu.se
Atakhani and Bogdziewiez contributed equally to this work.

## Abstract

Cell–cell adhesion is a fundamental feature of multicellular organisms. To ensure multicellular integrity, adhesion needs to be tightly controlled and maintained. In plants, cell–cell adhesion remains poorly understood. Here, we argue that to be able to understand how cell–cell adhesion works in plants, we need to understand and quantitatively measure the mechanics behind it. We first introduce cell–cell adhesion in the context of multicellularity, briefly explain the notions of adhesion strength, work and energy and present the current knowledge concerning the mechanisms of cell–cell adhesion in plants. Because still relatively little is known in plants, we then turn to animals, but also algae, bacteria, yeast and fungi, and examine how adhesion works and how it can be quantitatively measured in these systems. From this, we explore how the mechanics of cell adhesion could be quantitatively characterised in plants, opening future perspectives for understanding plant multicellularity.

## 1. Multicellularity and cell-adhesion mechanics

All living organisms are exposed to physical stresses such as tension and compression, that are experienced during growth and development, or imposed by external stimuli. At the supracellular level, tension has been found to pull adjacent cells apart in mutant organisms in which the proper control of cell–cell adhesion is defective (Thomas et al., 2013; Verger et al., 2018). However, living organisms have evolved mechanisms to continuously respond and adapt to these mechanical forces to ensure the maintenance of their cellular and supracellular integrity (Hamant & Saunders, 2020; Hannezo & Heisenberg, 2019; Trinh et al., 2021). In turn, cell–cell adhesion not only enables adjacent cells to stick to each other in a passive manner, but it is also dynamically controlled and maintained over time during growth and development (Baum & Georgiou, 2011; Leckband & de Rooij, 2014; Yap et al., 2018). The precise fine-tuning of adhesion tightening and loosening ensures tissue cohesion and integrity while allowing cell migration (in animals) or, for example, intrusive cell growth (e.g., fibre cell elongation in wood formation; Gorshkova et al., 2012; Marsollier & Ingram, 2018), developmentally controlled organ abscission (e.g., leaf or petal shedding in plants; Olsson & Butenko, 2018) and cell separation (e.g., lateral root emergence, root cap shedding or intercellular space formation; Stoeckle et al., 2018) in plants. Understanding the mechanics of cell adhesion and how adhesion resists and adapts to separating forces is thus crucial to understand how multicellularity is successfully achieved through cell adhesion.

## 2. Cell-adhesion mechanics: strength, work and energy

The physics of adhesion is highly studied for a number of industrial applications, and some of this well-established knowledge can also be applied to the question of cell adhesion in living organisms (Creton & Ciccotti, 2016; Packham, 2017). When characterising adhesion mechanics, it is common to refer to the term adhesion 'strength', but adhesion 'work' and 'energy' are two other parameters describing the adhesion mechanics that should also be considered. A first estimation of adhesion mechanics (strength and work) can be obtained by simply pulling apart the adhesive cells and measuring the force required to separate them. Strictly speaking, it is

**A** Force, Stress, Strain

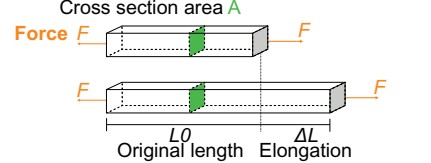

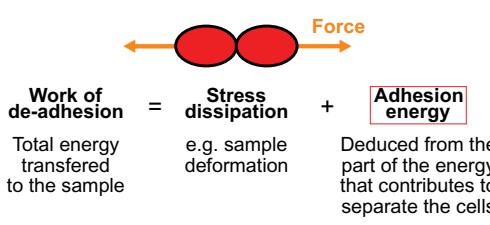

**B** De-adhesion strength and work     **C** Adhesion energy estimation

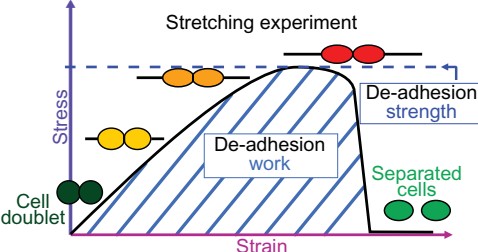

**Fig. 1.** Mechanics of cell adhesion. (a) Force, stress and strain. When a force (orange arrow) is applied on an object, it generates a mechanical stress and a strain (deformation). The stress corresponds to the ratio of the force $F$ applied to the cross-sectional area $A$ of the object on which the force is applied. The strain is the ratio of the elongation $\Delta L$ of the object to the original length $L_0$ of the object. (b) De-adhesion strength and work. When a doublet of cells is stretched apart, the cells are deformed (strain) and the stress in the sample increases until the stress is high enough to break the links between the cells. The maximal amount of stress applied before the cells detached from each other corresponds to the de-adhesion strength (blue dashed line). The area under the stress–strain curve is measured as the de-adhesion work (blue hatched area). After separation, the cell shape can be changed due to their plasticity. (c) Adhesion energy can in principle be deduced from the work of de-adhesion by considering stress dissipation.

in fact not directly a measure of the adhesion; instead, adhesion mechanics is inferred from measuring the de-adhesion. When performing a de-adhesion experiment, both the amount of force applied and the deformation of the cells can be measured, extracting the relation of the exerted force due to the applied displacement. Using the geometry of the sample and the values of force and displacement, a stress–strain curve can be generated (Figure 1a,b). From this curve, both adhesion strength and work can be calculated. The de-adhesion strength is determined by the maximal (ultimate) stress that was applied before the cells separated (Figure 1b). De-adhesion work corresponds to the energy that is transferred to the sample during the stretching. It is a function of both the amount of force and displacement (pulling) and is thus calculated as the area under the stress–strain curve of the de-adhesion experiment (Figure 1b). By contrast, the adhesion energy, which corresponds to the force generated by the molecular bonds formed between the cells, is not directly measurable with these approaches (Maître & Heisenberg, 2011; Sackmann & Smith, 2014; Winklbauer, 2015). Although it is in principle possible to infer the de-adhesion energy from a pulling experiment (Figure 1c), when energy is transferred to the sample by stretching, much of this energy can be dissipated in elastic and plastic deformation (extension) of the sample. The energy is either temporarily stored in the elastic deformation, or consumed in breaking molecular bonds within the sample, creating irreversible extension rather than separation at the cell–cell interface (Arroyo & Trepat, 2017). Other factors such as cell geometry and pre-existing stress can further contribute positively or negatively in dissipating the stress (da Silva et al., 2018; Lenne et al., 2021). It is thus complex to separate the effect of stress dissipation and infer the underlying de-adhesion energy and further the adhesion energy. Models from soft matter physics of adhesion have been proposed to predict adhesion energy in living cells, where cells are approximated to solid homogeneous spheres (Brochard-Wyart & de Gennes, 2009; Chu et al., 2005).

Because we know very little about the mechanics of cell–cell adhesion in plants, in this review, our aim is to explore our knowledge of cell adhesion in general and hypothesise on how the mechanics of adhesion could be quantitatively characterised in plants.

## 3. Cell–cell adhesion in plants

Adhesion between plant cells is mediated by their cell walls (Figure 2; Bou Daher & Braybrook, 2015; Jarvis et al., 2003; Knox, 1992). While adhesion is first established during cell division, each cell then synthesises its own cell wall through the secretion of polysaccharides (cellulose, hemicelluloses and pectins) and structural glycoproteins (Anderson & Kieber, 2020), creating a layered structure with a pectin-rich middle lamella between the walls of adjacent cells (Zamil & Geitmann, 2017). Beyond the middle lamella, tricellular junction and the edges of partially separated cells may represent mechanical hotspots in which mechanical stress is focused and adhesion is specifically reinforced (Jarvis, 1998; Roland, 1978; Willats et al., 2001). On the other hand, while strategically located, the contribution of plasmodesmata (intercellular channels) for cell–cell adhesion strength remains to be characterised. Overall, the cross-linking of the cell wall polysaccharides creates a continuous polysaccharide network that ensures cell–cell adhesion (Figure 2; Anderson & Kieber, 2020). Nevertheless, pectins, which are the main constituents of the middle lamella, are generally considered to be the main determinant of cell adhesion in the early stage of tissue growth and development (Bou Daher & Braybrook, 2015). Pectins are, in fact, a complex set of polysaccharides mainly composed of homogalacturonans (HGs), rhamnogalacturonans I, and rhamnogalacturonans II (RGIIs; Anderson, 2019). The HG is a linear chain of galacturonic acids (GalAs) that is

synthesised in a highly methyl esterified form, and that can be de-esterified after secretion to the cell wall by proteins called pectin methylesterases (Hocq et al., 2017). The de-esterification process leaves negatively charged residues on the GalAs, and if more than 9 consecutive de-esterified GalAs are present, they can form calcium-mediated bridges with another de-esterified HG under the so-called egg-box conformation, effectively cross-linking independent HG chains potentially coming from adjacent cells (Huxham et al., 1999; Jarvis & Apperley, 1995; Liners et al., 1989; Willats et al., 2001). Similarly, RGII, a very complex polysaccharide (Bar-Peled et al., 2012), also allows cross-linking of polymers through dimerisation mediated by the presence of Boron (O'Neill et al., 2001). The importance of these types of cross-linking for cell adhesion has been supported by the characterisation of cell-adhesion defective mutants. For instance, in Arabidopsis, several mutations in genes involved in HG synthesis (Bouton et al., 2002; Du et al., 2020; Lathe et al., 2021; Mouille et al., 2007; Temple et al., 2021), remodelling (Francis et al., 2006; Lionetti et al., 2015) or degradation (Ogawa et al., 2009; Rhee et al., 2003; Rui et al., 2019) harbour cell-adhesion phenotypes (loss of cell adhesion or incapacity of cell separation). Furthermore, selectively digesting the cell wall with pectin degrading enzymes, such as polygalacturonases, and even chelating the cell wall calcium with chelators, such as EDTA, can also lead to cell separation (Letham, 1960; McCartney & Knox, 2002; Ng et al., 2000; Zhang et al., 2000). However, not all cell types can be separated with these approaches (McCartney & Knox, 2002). Different tissues, cell types and even cell faces have significantly different cell wall compositions, and adhesion in those different contexts may rely on a different combination of polysaccharides and cross-links. Other work suggests that these are not the only molecular links that are responsible for cell adhesion and ferulic acids (Ng et al., 1997) as well as xyloglucan-like polysaccharide may, for instance, be involved (Ikegaya et al., 2008; Ordaz-Ortiz et al., 2009). Furthermore, during abscission events, many cell wall remodelling enzymes, other than pectin degrading enzymes, get expressed and likely act in concert with the pectin degrading enzymes to promote cell separation (Cai & Lashbrook, 2008; Lashbrook & Cai, 2008; Meir et al., 2010). The recent isolation and characterisation of the suppressor of an Arabidopsis mutant deficient in HG and showing cell-adhesion defect also revealed that adhesion could be restored without directly compensating for the HG deficiency in the cell wall (Verger et al., 2016). The loss of cell adhesion in these HG-deficient mutants may instead be largely due to secondary responses to cell wall integrity defects and further cell wall degradation (Du et al., 2020). Thus, other components and cross-links in the primary cell wall likely also play an important role in cell–cell adhesion. Although much less described, when transitioning to secondary cell wall formation, the pectins appear to be largely replaced by lignins which in turn take up the role of keeping the cells attached (Ciesielski et al., 2014; Li & Chapple, 2010; Yang et al., 2020). Similarly, other plant species, like most grass, have cell walls that contain very small amounts of pectins, such that adhesion should rely on different types of cross-links in these plants. Overall, a number of studies have started to investigate the chemical basis of cell adhesion. Furthermore, many of the cell wall properties and modifications identified as playing a role in adhesion have been correlated in other studies with specific mechanical properties (Grones et al., 2019; Zhang et al., 2019). For instance, de-esterification of the HG can lead to a stiffening of the cell wall due to increased calcium mediated cross-linking and could thus correlate with a stronger adhesion (Willats et al., 2001). On the

other hand, pectin de-esterification has also been associated with cell wall softening (Peaucelle et al., 2011; Wang et al., 2020). This could be a consequence of a higher susceptibility of de-esterified HG to pectin degrading enzymes [e.g., polygalacturonases (PGs)] or result from electrostatic repulsion of the carboxyl groups (Cosgrove & Anderson, 2020; Haas et al., 2020; 2021; Wang et al., 2020). Thus, while pectin digestion by PGs would lead to cell separation, cell wall softening without HG degradation could instead make the cell–cell connection more elastic and resilient to external stress and strain (da Silva et al., 2018). Similarly, while increased calcium-mediated cross-linking is intuitively expected to make adhesion stronger, the matrix stiffening could instead make the cell–cell interface more brittle and increase the risk of cell wall fracture and cell separation. However, at this point, it remains very difficult to predict how chemical modifications of the cell wall affect the mechanical properties of the cell wall (Zhang et al., 2016) and furthermore how the mechanical properties of the cell wall affect adhesion strength, work and energy.

Beyond direct cell wall regulators, a number of molecular players are known to be involved in the regulation of cell-adhesion maintenance and developmentally regulated cell separation. So far, most interest has been given to the signalling events leading to cell separation (Olsson & Butenko, 2018). Briefly, patterning transcription factors define abscission and dehiscence zones (Mao et al., 2000), and auxin and ethylene act, respectively, as inhibitors and activators of the abscission process (Meir et al., 2010), which is then regulated by small peptides and receptor-like kinase signalling (Stenvik et al., 2008). This triggers a mitogen-activated protein (MAP) kinase signalling cascade which ultimately activates transcription factors that promote the expression of cell wall remodelling enzymes leading to cell wall digestion and cell separation (Shi et al., 2011). On the other hand, much less is known about the molecular mechanisms that contribute to maintaining cell adhesion during growth and development, but the isolation of various cell-adhesion defective mutants brings some clues. For instance, in Arabidopsis, mutants defective in actin filament nucleation (Le et al., 2003; Mathur et al., 2003a; 2003b; Qiu et al., 2002; Saedler et al., 2004), a putative mechanosensitive calcium channel (Galletti et al., 2015; Tran et al., 2017) and epidermal identity transcription factors (Abe et al., 2003), show cell-adhesion defects. Interestingly, these mutants hint at a complex regulation of cell-adhesion maintenance through mechanical feedback as well as developmental control. Much remains to be clarified about the actual mechanochemical basis of cell–cell adhesion in plants. On the other hand, our understanding of cell adhesion in animals is much more advanced in part due to the existence of quantitative micromechanical methods to measure cell-adhesion mechanics.

## 4. Cell adhesion in animals, fungi, bacteria and algae

In animals, cell–cell and cell–substrate adhesion is primarily mediated by a group of proteins called cell-adhesion molecules (CAMs), that includes the cadherins (Halbleib & Nelson, 2006; Leckband & de Rooij, 2014) and integrins (Iwamoto & Calderwood, 2015; Lodish et al., 2008; Roberts et al., 2002). The CAMs are transmembrane proteins that harbour an extracellular domain at the surface of the cell, that can either bind to another CAM, the extracellular matrix or a substrate (Figure 2). They organise in patches called cell junctions that are responsible for cell–cell and cell–substrate adhesion, and mediate the dynamics of motile cells. During cell–cell adhesion, the cadherins also interact with the actin network

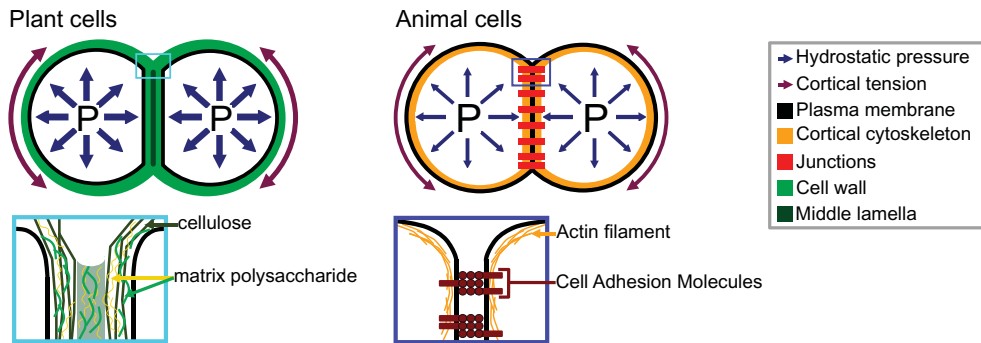

**Fig. 2.** Adhesion in plant and animal cells. The top drawings represent cell doublets, and the bottom drawings are close-up representations of the edge of the cell–cell interface. In both plant (left) and animal (right) cells, the cell shape is governed by an equilibrium between the internal hydrostatic pressure and the cortical tension. The plant cell adhesion is mediated by a cell wall mostly composed of cellulose and matrix polysaccharides. At the interface, the middle lamella is enriched in pectins and believed to play an important role in adhesion. The animal cell adhesion is mediated via proteins located at the plasma membrane. Cell-adhesion molecules are linked to the actin cytoskeleton which contributes to the cortical tension.

through the catenins and vinculin (Halbleib & Nelson, 2006; Harris & Tepass, 2010; Hartsock & Nelson, 2008). Together they form a mechanosensing complex sensitive to intrinsic (actin/myosin contractility) and extrinsic mechanical signals (Yap et al., 2018). Any force in the range of 5 pN, such as the force generated by the movement of myosin II motor in the cytoskeleton, can cause a conformational change of the $\alpha$-catenin and increase the affinity to vinculin. The recruited vinculin promotes actin nucleation and the recruitment of additional cadherins, which changes the cell's mechanical properties and cell-adhesion strength (Pinheiro & Bellaïche, 2018; Yap et al., 2018). But comparing the cadherin binding energy ($5 \times 10^{-20}$ J per cadherin) to the cell–cell binding energy (0.02–4.1 mJ/m$^2$) reveals the need for essential factors to justify the adhesion strength as the role of cadherin–cadherin junctions are less than 30%. The cadherin–cadherin adhesion tension is just enough to create contacts between the cells. Studying the cell junctions showed that the adhesion strength is either due to an increased overall cortical tension or the increase in cadherin density in the cortex (Winklbauer, 2015). In parallel to cell–cell adhesion, during cell–substrate adhesion, integrins interact with the substrate as well as with actin network through the talin and vinculin (Zamir & Geiger, 2013). As for cell–cell adhesion, tension induces adhesion strengthening leading to the formation of focal adhesions sites at the end of so-called actin stress fibres (Haase et al. 2014; Harris et al., 2012; Kuo, 2014; Rosen & Dube, 2006). For more detailed information on the molecular mechanism of adhesion in animals, see Garcia and Gallant (2003), Harris and Tepass (2010), Leckband and de Rooij (2014) and Pinheiro and Bellaïche (2018).

Further away from the animal cells are fungi, bacteria and algae. Because of their cell wall, cell adhesion in these organisms may be more similar to plant cells. These organisms are found in the form of unicellular biofilms and multicellular organisms depending on the environmental conditions and species (Carpentier, 2014; de Groot et al., 2013; Hall-Stoodley et al., 2004; Nagy et al., 2018; Raven & Giordano, 2014; Sheng et al., 2007). In many of these organisms, cell–substrate adhesion has been intensively studied. In general, it is a two-stage process, starting with a passive adhesion followed by an active adhesion. The passive adhesion is the first stage of the attachment, and it is reversible (Jones, 1994; Lebret et al., 2009; Razatos et al., 1998). This is mediated by the pre-existing cell wall or extracellular matrix and happens due to the existence of Van der Waals Forces, short-range hydrophobic interactions, electrostatic interactions and hydrogen bonding between the cell wall and the substrate. In the active adhesion stage, the organisms further generate a glue-like substance in aqueous environments to enhance the adherence to the substrate or the host cell. There is a large diversity of components in the cell wall and the extracellular matrix of living organisms that can bind to and alter the substrate. This creates strong anchorage between an organism and its substrate or host cells (Epstein & Nicholson, 2016; Kostakioti et al., 2013). For further information on the molecular mechanism of adhesion in fungi, bacteria and algae, see Nagy et al. (2018), Raven and Giordano (2014) and Tuson and Weibel (2013).

Overall, research on the characterisation of cell–cell adhesion mechanics is already quite advanced concerning animal cells. Similarly, much research on mechanical characterisation of cell–substrate adhesion have been done in animals, bacteria, yeast, algae and fungi. On the other hand, mechanical and quantitative characterisation of cell adhesion in plants is largely lacking to date. The mechanics of adhesion is also different in principle from that of animal cells, since in plants and other walled organisms, adhesion is mediated by their large cell wall, whereas in animals, it is mediated by clusters of adhesion proteins located at the cell surface. During the de-adhesion process, walled cells may separate from crack initiation in the cell wall and propagation in a manner similar to a crack in a hydrogel (Arroyo & Trepat, 2017; Creton & Ciccotti, 2016) rather than by the dissociation of CAMs (proteins, e.g., cadherins). In plants, a distinct cell wall layer, the middle lamella (Zamil & Geitmann, 2017), plays the role of interface between adjacent cells, and when cells separate, such cracks may preferentially propagate along the middle lamella when it is weakened. Furthermore, compared to animals, adhesion in plants appears as a less dynamics process because it is mediated by their rigid cell wall. But adhesion in plants is not static as cells can detach, reattach and grow intrusively. Interestingly, the active stage of cell–substrate adhesion described for yeast, algae and fungi could share functional similarities with the mechanisms of cell–cell adhesion maintenance in plant or the mechanisms allowing plant grafting. In turn, unicellular green algae with cell wall composition similar to that of embryophytes (true plants) have been proposed as models to understand plant cell–cell adhesion (Domozych et al., 2007, 2014). As such, the methods applied to study cell–

substrate adhesion in these 'walled' organisms could also be useful to understand cell–cell adhesion in plants.

## 5. Methods to quantify cell adhesion

### 5.1. Quantifying large-scale cell–substrate adhesion

Most of the early adhesion measurement methods that have been developed focused on measuring cell–substrate adhesion. They are based on applying shear stress on cells thanks to fluid flow or centrifugation and study a large number of cells at once to obtain a relative value for adhesion strength (i.e., percentage of cells detached by a given force). The plate-and-wash assay was the first quantitative method developed to measure cell-adhesion strength. In this method, the cells are placed in contact with a substrate for a certain time in order to let them adhere to the substrate (Klebe, 1974). Then a washing step is executed by removing the growth medium and replacing it with a new one. The flow created by the removal and addition of the medium creates shear stress that leads to the detachment of the least adhesive cells (Figure 3a). The number of cells attached is thus counted before and after this washing step. This technique gives a first idea of the adhesion between a cell type and a substrate; however, it lacks sensitivity, and the applied forces are weak and very difficult to control (Alam et al., 2019). To contravene these limitations, more advanced fluidic-based assays have been developed (Figure 3). In the centrifugation assay, the sample and the substrate are placed in 96-well plates allowing multiple measurements at the same time. A controlled detachment force, perpendicular to the cell–substrate contact area, is applied directly on the cells removing the nonadherent cells from the substrate surface (Figure 3b). Again, the number of cells is counted before and after the application of a force which in this case can be calculated more precisely based on the properties of the centrifuge. In another type of approach based on hydrodynamic flow, a fluid flow passing in a channel over a large population of adherent cells creates a detachment shear force (Figure 3c; Garcia & Gallant, 2003). In this case, the shear stress can also be determined precisely, and depends on the cross-sectional area of the channel containing the cells and through which the fluid is passing, the viscosity of the fluid and the flow rate (Ungai-Salánki et al., 2019). Several shapes of flow chambers exist like the radial flow chamber, the spinning disc or the parallel plate assay (Figure 3c; García et al., 1997; Kooten et al., 1992; Ungai-Salánki et al., 2019). More recently, microfluidic devices have replaced these flow chambers. Micrometer range channels ensure a laminar flow, and the microfabrication allows complex designs with multiple channels of increasing sizes which allows for a large range of fluid velocities, including high shear stress (Figure 3d; Christ et al., 2010; Lu et al., 2004). This is a major improvement for this approach since many cell-adhesion measurement techniques are limited by the amount of force that can be applied to the cells. Overall, these methods have been proved very useful to understand cell–substrate adhesion and have been largely improved in terms of accuracy, but they are designed and often limited to the study of cell–substrate adhesion rather than cell–cell adhesion.

### 5.2. Cell shape-based adhesion energy estimation

Computational imaging techniques can be used to study the cell–cell or cell–substrate adhesion mechanics in a noninvasive manner (Veldhuis et al., 2017). They are based on studying the mechanical equilibrium between the cell's cortical tension and adhesion (Figure 3e). More precisely, the cell's hydrostatic pressure creates a cortical tension at the surface of the cell, such that a single cell tends to minimise its surface tension and therefore becomes spherical (Winklbauer, 2015). However, when two cells are attached to each other or a cell adheres to a substrate, a flat contact interface is formed where the adhesion takes place, which goes against minimising the cell's surface tension. In this case, the adhesion energy is high enough to counteract the surface tension that would make the cell round. In turn, knowing the cortical tension and cell geometry, it is possible to deduce the adhesion energy when assuming the cells generally behave as fluids. This approach can be extended to a tissue where the cells still minimise their surface tension but are adherent to other cells and similarly the shape of the cells will then depend on an equilibrium of the adhesion and the cortical tension (Manning et al., 2010; Veldhuis et al., 2017; Winklbauer, 2015). Interestingly, with this approach and contrary to the other approaches that measure de-adhesion, it is directly the cell-adhesion energy that is predicted. However current methods depend on the assumption that cells behave like fluids.

### 5.3. Micromanipulation assays

Another range of techniques employs direct and precise cell micromanipulation, to grab, put cells in contact and pull them apart or from a substrate and measure the force necessary to do so. These uses either microaspiration of the cells with micropipettes or attachment of a cell to a cantilever. The dual pipette aspiration method uses a pair of micropipettes that can each grab a single cell by microaspiration. The two cells can then be placed in contact with each other for a given time to establish adhesion and then pulled apart to measure the adhesion that has developed (Sung et al., 1986), while the manipulation takes place under a microscope to monitor the detachment of the cells. In recent implementations, the microaspiration is precisely controlled by microfluidic pumps providing negative pressure (suction) and the pipettes are held by motorised micromanipulators (Biro & Maître, 2015). Adhesion strength is then measured using the step-pressure assay (Figure 3h). Once the cells adhere to each other, one micropipette is moved backwards to pull the cells apart. At low suction in the measuring pipette, the cell held by the measuring pipette will detach from it rather than from the other cell. In the next step, the measuring pipette is brought back to the cell, the suction inside the measuring micropipette is increased and the cells are pulled again. These cycles continue until the suction is strong enough to separate the cells. The minimal suction needed to detach the cells is then used to calculate a value for the de-adhesion strength. Another approach called single-cell force spectroscopy uses an atomic force microscope (AFM) to quantify cell de-adhesion strength and work. In single-cell force spectroscopy (SCFS), a cell is attached to an AFM cantilever (without tip) and then placed in contact with a substrate (tissue, cell or functionalised surface; Benoit et al., 2000; Figure 3f). When attempting to detach the cell, the cantilever is first bent as the pulling force increases until the force is high enough to detach the cell and the cantilever then regains its original shape. The AFM measures this bending by quantifying the deflection of a laser pointed at the cantilever. In turn, knowing the spring constant of the cantilever, it is possible to calculate the force applied throughout the pulling experiment and derive a stress–strain curve from which the de-adhesion strength and work can be precisely calculated. The main difficulty of the SCFS is the necessity to coat the cantilever to attach the cell on it since the cell-cantilever adhesion must be higher than the cell–cell or cell–substrate adhesion and the

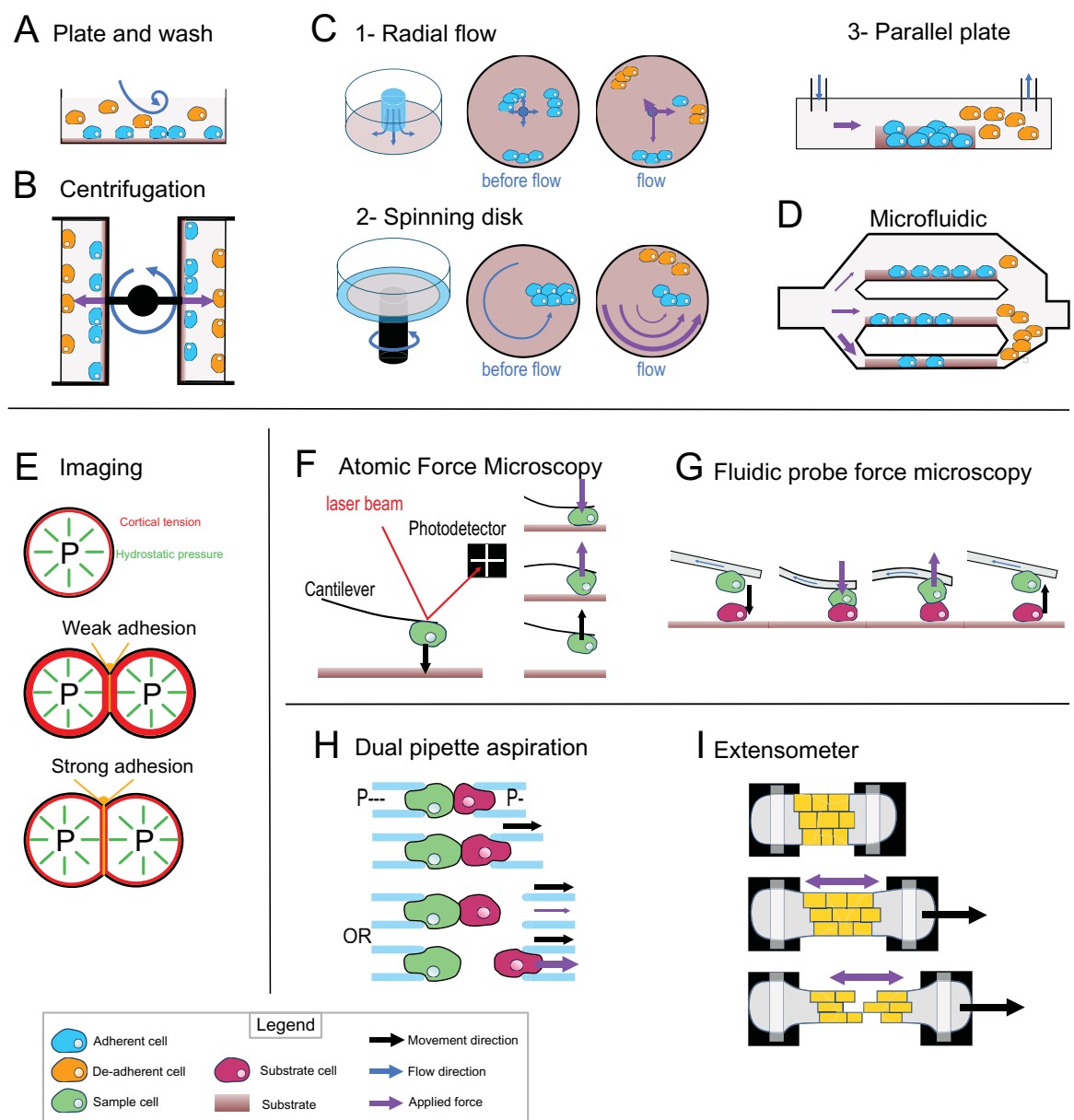

**Fig. 3.** Cell-adhesion quantification methods. (a) Plate and wash assay. After placing the cells in contact with the substrate for a period, the medium is replaced, removing the nonadherent cells. (b) Centrifugal assay. A 96-well plate is coated with a substrate. A cell suspension is added and allowed to adhere to the substrate. Then the wells are sealed, and the plate is spun at a specific speed and duration. (c) Hydrodynamic flow assays. (c1) The radial flow chamber. An outward radial flow. Different intensities of shear stress are applied on the cells depending on their position in the chamber according to a gradient inversely proportional to the radial position. (c2) The spinning disc assay uses a rotating flow to apply a hydrodynamic shear force on the sample. The cells are in contact with a substrate on a plate, which is rotating in a chamber containing liquid. The applied forces vary linearly with radial distance. (c3) The parallel plate. The cells are in a rectangular chamber, and a constant flow is maintained during one experiment. (d) Microfluidic chamber. The cells are in micrometre-range channels where a laminar flow is running. The smaller the channel is, the higher is the shear stress applied on the sample. (e) Cell shape-based imaging technique. The cell shape depends on an equilibrium between the internal hydrostatic pressure and the cortical tension. The strength of cell–cell adhesion can be deduced from the area of contact and the angle at the junction of cells. (f) Single-cell force spectroscopy. A cell is attached to the cantilever of an atomic force microscope then put in contact with a substrate. The cantilever first approaches and pushes the cell on the substrate. Then the cell is pulled from the surface. The adhesion force between the cell and the substrate will bend the cantilever until the cell detaches. A laser is reflected by the cantilever to a photodetector. This will draw a force curve giving information on the force of adhesion between the cell and the substrate. (g) Fluidic probe force microscopy. The same principle as the single-cell force spectroscopy, but the cantilever contains a microchannel that can suck the cell like a micropipette. (h) Step pressure assay. Two motorised micropipettes are connected to a microfluidic pump. First, the two cells are in contact to allow the adhesion. One micropipette keeps the cell in position with a high constant suction, whereas the other one pulls its cell away with a lower suction. If the adhesion is stronger than the measuring pipette suction, the cell will slip out the micropipette. The suction in this pipette is then increased until it is strong enough to separate the cells. (i) Extensometer used for tensile test. The tissue sample is held at each end and stretched with a specific force. The stretching and the tearing are monitored with microscopic observation. For additional information on the methods presented, including the quantitative/mathematical framework used to derive adhesion mechanics values, see Alam et al. (2019), Garcia and Gallant (2003) and Ungai-Salánki et al. (2019).

coating can chemically disrupt the state of the cell and therefore modify its adhesion properties (Cohen et al., 2017; Guillaume-Gentil et al., 2014). This issue can be avoided by using fluidic probe

force microscopy (Meister et al., 2009). In this case, the cantilever contains a microchannel and can thus act as a micropipette, which sucks the cell so that no coating of the cantilever is needed to hold

the cell (Figure 3g). Overall, these micromanipulation techniques bring a much higher precision in the measurement of adhesion compared to the other methods described above but have a much lower throughput and are limited to the study of single cell–cell or cell–substrate adhesion.

### 5.4. Tissue-level measurements

Most of the methods existing to investigate the mechanics of cell adhesion are single-cell techniques. Measuring mechanical properties at the tissue level can be difficult because of the heterogeneity of the cells or extracellular material that compose a tissue, but a few methods exist. The 90° peel test method has been used on cell layers cultured on a substrate to measure the adhesion force of this cell layer to its substrate (Uesugi et al., 2013). Tensile tests have also been developed to explore the mechanical properties of a tissue and have been applied to measure cell adhesion in the tissue context. For such methods, the sample is placed in an extensometer, a device that allows the stretching of the samples and a measurement of the load applied during stretching (Figure 3i). The sample is attached at its extremities and is then stretched while the deformations and cell detachments are monitored with a microscope (Harris et al., 2012). Similarly, to the single-cell approaches, the tissue is stretched until the cells separate, and the de-adhesion strength is calculated locally from the observed strain and predicted stress during the stretching. While more complex to analyse, such a tissue-level approach brings more physiologically relevant information concerning cell–cell adhesion in the tissue context compared to single-cell approaches.

### 5.5. Cell-adhesion quantification methods in plants

While still very limited compared to other organisms, quantification of cell-adhesion mechanics has already been approached in plants. Following a method first established by Parker and Waldron (1995), Leboeuf et al. (2004) used 'vortex-induced cell separation' to study the differences in adhesion between *Arabidopsis thaliana* microcalli suspension cultures at several time points during growth. The method consists of measuring with a particle size analyser the size of cell clusters from in vitro cell cultures before and after vortexing at 40 Hz. However, this method remains rather imprecise, the force applied to detach the cells is not measured and it is not entirely clear whether cell clusters get fragmented because of cell detachments or cell ruptures. Tensile tests have also been used on plant samples to explore cell-adhesion establishment after grafting (Kawakatsu et al., 2020; Lindsay et al., 1974; Moore, 1984; Notaguchi et al., 2020), but again it remains difficult to determine if cells are being separated or if tissues rupture. Melnyk et al. (2018) used an automated confocal micro-extensometer (ACME; Robinson et al., 2017) in such experiments to determine grafting adhesion. Since this device is coupled with a confocal microscope which allows live 3D imaging during the stretching, it revealed that indeed most cells at the grafting interface appear to break rather than separate when the sample is stretch until rupture, suggesting that cell adhesion of previously separated cells becomes stronger than the cohesive strength of the cell cortex (cell wall). Tensile tests have also been used at the level of the cell wall, in strips of microdissected onion outer epidermal cell wall containing cell–cell junctions (Zamil et al., 2014). Interestingly, it also revealed that the cell walls did not separate at the cell–cell junction but rather broke within the cell cortex. Another model of study of adhesion

in plants is the pollen grain and its adhesion to a substrate or to the stigma. Jauh et al. (1997) developed a method similar to the plate and wash assay to measure pollen adhesion on an artificial matrix. They blotted exudates from Lily (*Lilium sp.*) styles on nitrocellulose membranes and tested the adhesion of pollen grains after gently washing with a liquid buffer. Similarly, Zinkl et al. (1999) used a plate and wash-like liquid assay. They directly used unpollinated pistils, pollinated them and washed away the nonadherent pollen with a liquid buffer. While this method still lacks precision in terms of the force applied, they also developed a high-precision approach that resembles single-cell force spectroscopy to study pollen adhesion strength of a single pollen grain to the stigma. This system has similarities with the force measurement principle of an AFM but directly uses pistils to bind the pollen grain from each side. One pistil is maintained on a translation stage and serves as a substrate. Another pistil is glued to a glass fibre cantilever. A pollen grain is then placed in between the two stigmas, and the translation stage is moved away from the cantilever, while the bending of the cantilever is measured via a reflecting laser beam. When the force is too high for one of the pollen adhesion sites, the cantilever retracts and the adhesion strength can be determined. Finally, other researchers have approached the study of plant cell-adhesion mechanics form the modelling side by developing analytical models (Jarvis, 1998) and finite element simulations of the middle lamella (Shafayet Zamil et al., 2017) as well as de-adhesion mechanics (Diels et al., 2019; Liedekerke et al., 2011) that were then compared with experimental data (Diels et al., 2019). Overall, while these approaches have initiated the investigation of cell-adhesion mechanics measurements in plants, they are still either limited to a specific case (pollen, grafting), remain very imprecise in terms of the actual forces applied (vortex) or are theoretical (modelling). In the remainder of this review, we thus want to explore and hypothesise on the methods that could be implemented to precisely quantify cell–cell adhesion mechanics in plants.

## 6. Implementing cell-adhesion measurement methods in plants

Studying cell adhesion in plants brings additional challenges. Plants have large and relatively rigid cell walls which are under high tensions due to the high turgor pressure of the cells. Plants cells (with the exception of specific cases like pollen grains) do not seem to adhere specifically to substrates contrary to most animals, bacteria, algae and fungi. Furthermore, here our focus is to understand cell–cell adhesion in the context of multicellularity rather than to study how a plant cell would attach to a substrate. Nevertheless, many of the approaches presented above could in principle be adapted to study plant cell adhesion, including the cell–substrate adhesion quantification methods. As explained above, some versions of the plate and wash assay have already been applied to the study of pollen grain adhesion, but they will likely not be as straightforward to apply to other plant cell types. Similarly, important adaptations may be needed for hydrodynamic flow approaches. Indeed, in order to isolate single plant cells, it is necessary to first remove them from their tissue by digesting their cell wall. The protoplasts obtained are very fragile and lack the cell wall that is the structure that mediates cell adhesion in a physiological context. It is thus necessary to first wait for the cell wall to regenerate (Pasternak et al., 2021). The adhesion of most plant cell types is actually cell–cell rather than cell–substrate, so another major adaptation will be the substrate to be used. A first possibility would be to use a live sample or a cell

layer such as the inner side of an epidermal peel as a substrate for an approach similar to a plate and wash assay. However, this still brings the limitation of not being able to measure the actual force imposed on the cells during the washing step and may not be applicable to flow and microfluidic chamber approaches. Yet, it could still be a very useful approach to quantitatively compare adhesion of wild type and mutant cells or in response to treatments. An alternative would be to mimic an adjacent cell wall using purified cell wall components such as pectins, celluloses and hemicelluloses, cell wall extracts blotted on a nitrocellulose membrane or coated in flow or microfluidic chambers or even other artificial scaffolds as recently developed (Calcutt et al., 2021). While it remains to be determined whether such study of plant cell adhesion to a substrate would provide physiologically relevant information, using different substrates with different composition could be very informative to further understand the chemical determinants of cell adhesion and similarly to compare wild type, mutants and treatments in a context in which the force applied on a large number of cells can be precisely quantified.

The computational imaging-based techniques would be at first glance the easiest to apply to plant cells. Starting from isolated cells, it is possible to let the single cells divide and spontaneously form cell doublets. One important limitation, however, is that it is unlikely that we can consider plant cells with a regenerated cell wall to behave as liquid. In turn, the calculation of adhesion energy based on the shape and cortical tension of the cell doublet may not directly apply here. However, the observation of cell contact shape in wild type, mutants and treated samples could still be very informative (Calcutt et al., 2021). The cell doublets could be trapped in microfluidics chambers allowing time-lapse live observation while changing the osmolarity of the medium and thus the turgor pressure and cell wall tensions. Increases in cortical cell wall tension could in principle force the cells into more spherical shapes and induce partial separations. Using finite element simulation of the system and considering measurable physical properties, such as cell wall elasticity, plasticity, thickness, turgor pressure, cell shapes and strain, following the changes of pressure, could generate an informative prediction of the local stresses in the cell wall and at the cell–cell interface. This would bring the possibility to deduce the cell–cell adhesion strength of cells that would partially detach as a consequence of changes in tension (Diels et al., 2019; Jarvis, 1998). While still far from a physiological context, such approach would allow the study of actual cell–cell adhesion of the cell doublet and in a minimally invasive system. One limitation, however, may be that in many cases, including untreated wild-type cells, the tensions induced by the changes in pressure may not be sufficient to separate the cells, or that they would only induce cell wall extension rather than separation of the interface. Conversely, micromanipulation techniques could be used to directly apply separating forces to cell doublets until the cells would separate. The step pressure assay with motorised dual micropipette could be used to attempt to separate spontaneously generated cell doublets or be used to put in contact single plant cells and measure the cell–cell adhesion strength developing over time. Similarly, the fluidic probe-based single-cell force spectroscopy approaches could in principle be used to measure the adhesion of a plant cell to a synthetic cell wall-like substrate. While the suction developed by the currently available micropipette-based setups may not be sufficient to separate the potentially very strong plant cell–cell adhesion (e.g., in the case of wild-type untreated spontaneously generated cell doublet), new implementations using micropipettes with larger diameters adapted to the size of plant cells and high negative pressure fluidic

pump could be developed to attempt to solve this issue. It remains that, as previously shown (Melnyk et al., 2018; Zamil et al., 2014), the cell cortex strength may often be weaker than the cell–cell adhesion. In this context, such a setup would ideally be able to develop pulling forces high enough to bring the cell to rupture and allow to assess the maximal measurable adhesion strength. On the other hand, at the tissue scales, stretching until rupture is already achievable in plants with a micro-extensometer. Similarly, to the case of grafting, a tissue sample can be firmly attached to opposite arms of an extensometer and stretched until it ruptures while measuring the force applied. The ACME setup allows the 3D live imaging of cell detachment in mutant samples (Robinson & Durand-Smet, 2020) and could be particularly useful to compare different cell-adhesion-defective mutants or the effect of different treatments. Here, the main limitation rather lies in the complexity of the sample. Most tissues are composed of multiple-cell layers, including the epidermis, cortex and the vasculature. While previous work has shown that the epidermis is put under tension by the extensometer stretching (Robinson & Kuhlemeier, 2018), it is not clear what is the load-bearing contribution of the different cell layers. In particular, the epidermis appears to be the first to yield and may at first be under the most tension, whereas the vasculature seems to be the last to yield. As such, quantifying the ultimate force and thus the strength of the samples will likely not really provide information regarding the properties of cell–cell adhesion in the samples, but rather on the toughness of the vasculature. In turn, a more complex analysis of the stress–strain curve in parallel with the microscopic observation with confocal microscopy (ACME) and coupled with computational mechanical simulation of the system (Diels et al., 2019) would be required to determine adhesion strength in the epidermis or cortex and compare different mutants or treatments. Finally, an alternative to this problem would be to study epidermal peels. While these may be particularly difficult to obtain from adhesion-defective mutants since the epidermal cells would easily detach from each other, this may be a very useful tool to study the effect of treatments or even study adhesion in transgenic lines in which cell-adhesion defect could be induced by the controlled expression of cell-separating enzymes or the controlled knockdown of genes involved in cell adhesion.

To summarise, from these ideas and hypotheses, much remains to be done and tested to determine if or which of these approaches will be the most useful and applicable for the mechanical quantification of cell adhesion in plants. We can hope that this will be the case for many of them, offering a range of techniques from relative measurement of the adhesion of large populations of cells to the very precise measurements in single-cell adhesion or adhesion in the tissue context. A successful approach could also be a combination of some of the approaches proposed (e.g., combining micromanipulation with finite element methods), or require the design and development of completely new approaches.

## 7. Conclusion

When it comes to multicellular organisms, we know adhesion is responsible for their integrity. We also have some knowledge on how this adhesion is mediated and kept through the organism's life span. In some organisms, like in animals, adhesion has been intensively studied at the single-cell level and molecular scale. Collaborations among biologists, physicists and engineers have led to the development of different devices and methods to determine

or measure the adhesion strength from the molecular to the tissue scale. But the focus of previous studies and methods developed has been rather on the single cell–substrate adhesion than multicellular cell–cell adhesion. The lack of methods for measuring cell–cell adhesion in plants is currently the main bottleneck for the field, and largely hampers our current understanding of the mechanisms controlling cell–cell adhesion in plants. As shown for the animal cells, methods to quantify cell adhesion have had a major impact to understand the molecular basis of cell adhesion but also the complex biophysical implications such as the fact that beyond cell–cell adhesion strength, the regulation of cortical tension, as well as mechanical feedbacks, plays a major role in controlling and maintaining adhesion (Lenne et al., 2021). Future development of precise quantitative methods for measuring cell adhesion in plants should thus open the door for major breakthroughs in the field and have a broad impact on our understanding of plant biomechanics.

## Acknowledgement

We are grateful to Rawen Ben-Malek and Grégory Mouille (IJPB, INRAE) for their helpful comments on the manuscript.

**Conflicts of interest.** The authors declare that no competing interests exist.

**Financial support.** This work was supported by grants from the Kempe Foundation (A.A.; grant number JCK-1912.2), Åforsk (L.B.; grant number 20.502) and the support of Bio4Energy (S.V.) and the UPSC Centre for Forest Biotechnology (L.B.).

**Authorship contributions.** All the authors wrote the manuscript. L.B. made the figures.

**Data availability statement.** No data or code was generated as part of this manuscript.

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
