## [Reviewer Report]

To the Editor, 

Please find attached our manuscript entitled “Characterizing the mechanics of cell-cell adhesion in plants” for Quantitative Plant Biology within the “call for paper: Plant biomechanics”. 

Adhesion is essential for keeping single-celled and multicellular organisms alive and developing. Despite being a fundamental feature, we still know very little about how it works in plants. Here, we argue that the lack of methods for quantitatively measuring cell-cell adhesion in plants is currently the main bottleneck for the field. Yet, adhesion mechanics in animal, fungi, yeast, algae and bacteria has been vastly studied thanks to the use of quantitative methods. Furthermore, the data generated have contributed to a large number of theoretical studies on the mechanisms of cell adhesion in these organisms. Here, we both intend to review the existing knowledge on cell adhesion mechanics in general and how it is quantitatively measured. A large focus is placed on presenting information on the tools and methods available for measuring the adhesion force at the single cell and the tissue level. We then hypothesize and propose possible future development of precise quantitative methods for measuring cell adhesion in plants. We believe this review is among the first steps in the development of such quantitative approaches for plants and that such developments will open the door for major breakthroughs in the field and have a broad impact on our understanding of plant biomechanics. 

Because the focus is on the development of biophysical quantitative methods, in our view, the collected material in this manuscript will be of interest to the readership of Quantitative Plant Biology. The knowledge reviewed has a high potential to serve as a reference in several scientific communities including development, biophysics and plant and animal biology in general, and the novel ideas proposed could fuel the development of new quantitative methods for plants biophysical characterization. Due to the far-reaching implications and the interdisciplinary nature of this review, we hope that you and the reviewers will consider this work for publication in this journal. 

Best regards, 

Stéphane Verger

---

## [Reviewer Report]

*Comments to Author*: This well written manuscript outlines the challenges to understand both the molecular mechanisms and the mechanics of cell adhesion in plants. Although principally focused on plants cells the review does consider what is known in other systems most notably animal cells and outlines a range of methodological approaches. The challenges of the study of cell adhesion for an organism with walled cells is well presented. The manuscript is thoughtful, authoritative and up-to-date and is well prepared and well-illustrated.

The manuscript is excellent in setting the scene for approaches – which may or may not be practicable for plant cells. The idea of preparing plant cell doublets for analyses is a good approach to think about - but of course may be less achievable in practice. The article is good in that it recognizes that cell to cell adhesion strength between plant cells may be greater than that of the cell wall – and thus any intervention to pull cells apart may lead to cell rupture rather than a controlled loss of cell adhesion.

The schematic representation of plant cells being rounded (as for animal cells) in Figure 2 – is perhaps not ideal and more defined shapes would indicate the importance of the cell wall in imposing cell geometries.

The authors could indicate understanding cell adhesion across plant organs is potentially even more challenging in that middle lamellae can have different origins – depending on, for example, their occurrence in transverse or in longitudinal walls in an elongating organ. The former being direct unexpanded regions deposited at cytokinesis and the latter arising from the often considerable extension of an original middle lamella/cell adhesion plane to be the interface between several cells due to directional cell elongation. Therefore it is possible, for example, that some treatments may release files of cells with transverse adhesions intact.

The authors are correct to highlight the lack of precise knowledge of how pectin functions in cell adhesion and they could also indicate that there is evidence in some cases that xyloglucan may be involved (Ikegaya et al. (2008) Presence of xyloglucan-like polysaccharide in Spirogyra and possible involvement in cell–cell attachment. Phycological Research 56, 216-222; Ordaz-Ortiz et al. (2009) Cell wall microstructure analysis implicates hemicellulose polysaccharides in cell adhesion in tomato fruit pericarp parenchyma. Molecular Plant 2, 910-921). There is also the issue of where the cell adhesion load is in relation to the complex polysaccharide networks across the two primary cell walls and middle lamellae that are between two adjacent cell membranes. What and where are the weakest links across such a cell interface and how are the molecules that have a primary function in cell adhesion linked into cell wall structures?

In summary, this is an excellent overview of the topic that should provide a keen stimulus for future research in this important area.

---

## [Reviewer Report]

*Comments to Author*: In this review article, the authors present the state of current knowledge of cell-cell adhesion in plants, as well as in animals and other species, introduce the mechanical concepts required to understand cell-cell adhesion, summarize a range of methods that have been used to measure cell-substrate or cell-cell adhesion in (mostly non-plant) biological systems, and discuss whether these methods could be used to measure cell-cell adhesion in plants and open new avenues of research. Cell-cell adhesion in plants is a fascinating and important topic that touches on many aspects of plant growth and development, and even on biotechnological aspects (for example, Yang et al. 2020, cited here, has implications for biofuel processing of poplar), but is poorly understood. Thus, this review article is timely and interesting, but could benefit from some revisions as suggested below.

Major Comments:

For a submission to Quantitative Plant Biology, I was struck by how non-quantitative the review is. Although it does a good job of presenting mechanics concepts in Figure 1, almost no numerical measurements or predictions of cell-cell adhesion forces in plants are reported, nor are they placed in a quantitative/mathematical framework that might allow for useful interpretation of experimental data, although finite element modeling is mentioned in passing. For example, in Line (L) 412, the authors posit that cell-cell adhesion is likely much stronger in plants than in animals; how much stronger, even to an order of magnitude?

Figure 1: Please standardize the font and font sizes throughout the figure (I suggest using 12 pt Arial for the panel letters and 10 or 8 pt Arial for all other text); panel A could be made larger to fill some of the white space and make it more readable. Since this review is mostly about plant cell adhesion, it might be helpful to depict more realistic cartoons of plant cells, making them rectangles or long hexagons rather than ovals.

Figure 2: Very few plant cell types are rounded like this, and cellular geometry likely influences cell adhesion mechanics, so I would suggest depicting both plant and animal cells more realistically here – as they stand now, the models remind me of the tendency in physics to approximate objects as spheres. Additionally, plant cells (and animal cells) are rarely present as doublets – I know that is for simplicity’s sake here, but maybe other surrounding cells could be shown in gray? These surrounding cells would influence cortical tension as shown. Junctions in the form of plasmodesmata are often present between plant cells, but are not shown here (or mentioned anywhere in the review) – I think they should be considered. I would also argue that we DO know something about cell wall organization (or at least composition) at the extremity of the cell-cell connection in plant cells, since researchers have detected particular structures/epitopes at tricellular junctions in various plant tissues (see, for example, work by Willats).

The section on animal cell-cell adhesion should be revised to more precisely define the molecular and cellular mechanics involved in cell adhesion. This section could also be connected to a new panel in Figure 3 that shows at least one example of these molecular players in a mechanical context.

The summary paragraph from L218-238 is repetitive and could be shortened – for example, the reintroduction of organ abscission or intrusive growth seems unnecessary. If you want to highlight these phenomena, why not show them in a figure, along with other examples of cell separation in plants such as pollen tetrad separation, pollen tube growth, and lateral root formation?

Section beginning L274: I am not sure cell shape-based adhesion energy estimation would apply to plant cells, given that they have semi-rigid walls that resist deformation; this caveat should be explicitly stated, rather than obliquely hinted at in the final sentence. This is covered later in the manuscript (L441), but these two ideas should be closely connected. In fact, it may aid the organization of the review to present each different method for measuring cell adhesion, then discuss how it could be applied to studying plants, rather than lumping all of the methods and all of the discussions together, separating each method from the discussion of how it could be applied (or not) in plants.

Cell-substrate adhesion has been studied in algae, such as Penium, that have wall composition similar to that of emrbyophytes (true plants); the authors could cite examples from Domozych and others of studies of Penium adhesion to substrates that is pectin-dependent.

Commas are often misused (mostly overused) in the manuscript. For example, in L30, the first two commas could be removed from the sentence starting “The precise fine-tuning”. In general, I would suggest reading the sentence out loud and adding commas only where one would naturally pause while speaking the sentence.

Minor Comments:

Many of the internal citations include first initials; these can likely be removed.

L24: Replace “But” with “However,”

L28: replace “rather” with “also”

L32: should be “fiber cell”

L34: “lateral” doesn’t need to be capitalized

L35: should be “space”

L36: should be “and how adhesion resists and adapts to”

L43: remove “the”

L50: From this sentence and the description in Figure 1, it is not clear how stress is independent of the dimensions of the sample – isn’t it defined as force over cross-sectional area? Please clarify – do you mean that the applied force and the strain are dimension-independent?

L85: I believe this citation should be “Bou Daher & Braybrook, 2015”, since Bou Daher is the first author’s surname

L118: remove comma

L128: I don’t think it is true that most monocots have small amounts of pectins in their cell walls, although grasses do. Grasses are a subset of monocots, so this sentence should be revised.

L157: should be “maintaining”

L159: would it be worthwhile to name these mutants, for instance in a table?

L166: remove “and it is”

L184: “and mediate”

L188: remove “a”; including a space between 5 and pN would aid readability

L192-196: please indicate the range of cadherin-cadherin adhesion tension and cell-cell binding energy to give the reader a quantitative sense of these values

L198: remove second “the”

L203: remove “the”

L207: revise “largely studied” to “studied in detail” or “intensively studied”

L215: the surface of what?

L216: revise to “This creates strong anchorage between an organism and its substrate or host cells”

L241: revise “rather focus” to “focused”

L243: remove “rather”

L254: should be “96-well”

L260: missing space between Fig. and 3C

L286: should be “96 well plate”

L302: write out “single cell force spectroscopy”

L303: could this be called a “suction pump” instead of a negative pressure pump?

L320: should be “use”

L329: why not consistently call this suction rather than negative pressure? Using both terms seems over-complicated

334: no need for the acronym SCFS since it is not very common and is only used three times in the main text

345: suggest replacing “denature” with “disrupt”; could use “adhesion properties” rather than “properties of adhesion”

L364: remove first comma

L366: “such a tissue”

L368: remove “the”

L370: “quantification of cell adhesion mechanics”

L374: “consists of”; remove comma

L375: replace “shaking by a vortex” with “vortexing”

L392: replace “Lilly stylar” with “lily (Lilium sp.) styles”

L402: they used a laser beam, not an ion beam, for that experiment

L420: “hydrodynamic”

L432: remove “plants”

L458: “micromanipulation”

L473: remove first comma

L477 “cell adhesion-defective”

L484: replace “inform of” to “provide information regarding”

L489: “treatments”

L506: replace “vastly” with “extensively”

L507: remove “The precious”

---

## [Reviewer Report]

*Comments to Author*: Both reviewers praised the quality of your manuscript. Reviewer 1 suggested a few changes and additions.They are mere suggestions, which you can decide to follow or not. Reviewer 2 asked for more revisions, although still minor, especially regarding the figures and the presentation of methods for measuring cell adhesion. These points should be addressed in your revised manuscript.